# A sequence-aware merger of genomic structural variations at population scale

Zeyu Zheng ®[1], Mingjia Zhu[1], Jin Zhang[1], Xinfeng Liu[1], Liqiang Hou[1], Wenyu Liu[1], Shuai Yuan[1], Changhong Luo[1], Xinhao Yao[1], Jianquan Liu ®[1] ✉ & Yongzhi Yang ®[1] ✉

Merging structural variations (SVs) at the population level presents a significant challenge, yet it is essential for conducting comprehensive genotypic analyses, especially in the era of pangenomics. Here, we introduce PanPop, a tool that utilizes an advanced sequence-aware SV merging algorithm to efficiently merge SVs of various types. We demonstrate that PanPop can merge and optimize the majority of multiallelic SVs into informative biallelic variants. We show its superior precision and lower rates of missing data compared to alternative software solutions. Our approach not only enables the filtering of SVs by leveraging multiple SV callers for enhanced accuracy but also facilitates the accurate merging of large-scale population SVs. These capabilities of PanPop will help to accelerate future SV-related studies.

Structural variations (SVs) are integral to genome evolution and function[1], encompassing a wide size range from 50 base pairs to megabases[2]. The emergence of advanced sequencing technologies has resulted in the generation of vast volumes of data, fueling the growing interest in SV research, especially within the dynamic landscape of the pangenomics era[3,4]. Amidst these research pursuits, a key challenge lies in the accurate identification of SVs in both individual genomes and entire populations, enabling the precise interpretation of SV-related information.

Several tools have been developed to detect structural variations (SVs), each possessing distinct advantages owing to their diverse methodologies[5–8]. While raw sequencing data theoretically hold information about all SVs, achieving accuracy necessitates a higher sequencing depth. Conversely, assembly data produce more precise results, but they may only capture SVs partially due to collapses or misassembly in heterozygosity region. Consequently, merging the outcomes of multiple SV callers proves advantageous, as it combines their strengths and yields a more accurate and comprehensive set of SVs for each individual. Furthermore, the investigation of SVs at the population level is gaining momentum, and the amalgamation of individual SVs into population-scale data is a crucial step[9–11]. Several software solutions are available for merging SVs based on position (Supplementary Table 1), such as SURVIVOR[12] and bcftools[13]. However,

these tools often encounter difficulties when handling different sequences of insertion/deletion in the same reference position, and they may mistakenly treat identical SVs in repeat regions as multiple SVs[14,15]. Recently, there have been advancements in software that consider both the sequence similarity and position of SVs, like Truvari[16], Jasmine[8], and SVanalyzer[13], which have greatly improved the accuracy of SVs merging. But when faced with SVs that partially overlap, these tools typically merge them into a single SV or treat them as complex multi-allelic SVs based on sequence similarity, which may result in the loss of important information or may be unsuitable for downstream analyses[15–17]. Furthermore, in this era of cluster computing, most of these software are designed for single-node and single-thread execution, which poses a challenge when dealing with a large number of samples for population SV merging. Therefore, achieving accuracy and efficient merging of SVs at the individual-scale with multiple callers or at the population scale remains a challenge[15].

To tackle these challenges, we have developed PanPop, a SVs merging toolkit that effectively handles combined SVs from diverse SV callers or multiple individuals. PanPop takes into account both the positional and sequence similarities of SVs and introduces an approach to splitting SVs based on sequence-aware alignment. This enables PanPop to effectively handle complex SVs, resulting in a greater number of bi-allelic variants available for further analyses. PanPop

[1]State Key Laboratory of Herbage Improvement and Grassland Agro-ecosystems, College of Ecology, Lanzhou University, Lanzhou, China. ✉e-mail: liujq@lzu.edu.cn; yangyz@lzu.edu.cn

exhibits high precision and delivers great performance in SV merging tasks, showcasing its versatility for both individual and population-scale datasets.

## Results

### Overview of PanPop pipeline

The central component of PanPop is a crucial process called PART (PAnpop Realign and Thin), which employs a sequence-aware SV local realignment method to resolve overlapping SVs, particularly reducing multi-allelic SVs into biallelic forms. PART consists of five steps: realign group, rebuild consensus sequences, alignment, SV integration, and SV thinning (Fig. 1a). During the realign group step, SVs that overlap or are in close proximity (with a default adjacency threshold of 200 bps) are grouped based on their positions. Subsequently, the rebuild sequences step retrieves the sequences of each realign group based on the mutation information in Variant Call Format (VCF) files and an SV parse process. During this process, if an individual contains multiple SVs within a realign group, each SV is treated as unique. The alignment process employs sequence aligners (MUSCLE[18], FAMSA[19], or stmsa[20]) to generate multiple sequence alignment results. MUSCLE is selected for short sequences (<1000 bp), while FAMSA or stmsa is used for longer sequences. The SV integration step involves dividing the alignment into distinct blocks along the reference sequence, where each block represents the unaltered status across different samples. Blocks displaying consistent alignment (without SVs) are eliminated, and only the remaining blocks are classified as new SVs. It is important to note that despite insertions of varying lengths altering the alignment status in different samples, they are classified as a single SV since we divide the alignment based on the reference sequence. Finally, the SV thinning process reduces multi-allelic SVs to biallelic ones based on their similarity. If two allele sequences share at least 60% of their bases and have a difference of less than 20 bp, they are considered the same allele. The remaining alleles are clustered into distinct groups using the MCL algorithm[21], with each group treated as a unique allele, and the alleles within each MCL group are merged together.

It is worth noting that PART is a relatively independent module specifically designed to handle the merging of different SV datasets in VCF format. All parameters can be easily and flexibly configured through the config file. In addition to utilizing PART for SV merging, the PanPop pipeline incorporates SV calling and filtering steps to integrate the entire process, starting from reads or assemblies and leading to the final individual or population SVs. For long sequencing reads, Minimap2[22] (or alternatively, NGMLR[23]) is employed to map them onto a reference genome. This is followed by four commonly used SV callers (Sniffles[23], cuteSV[5], swim[24], and pbsv) for SV identification. If an assembly is present, it is aligned using Minimap2, and SVs are

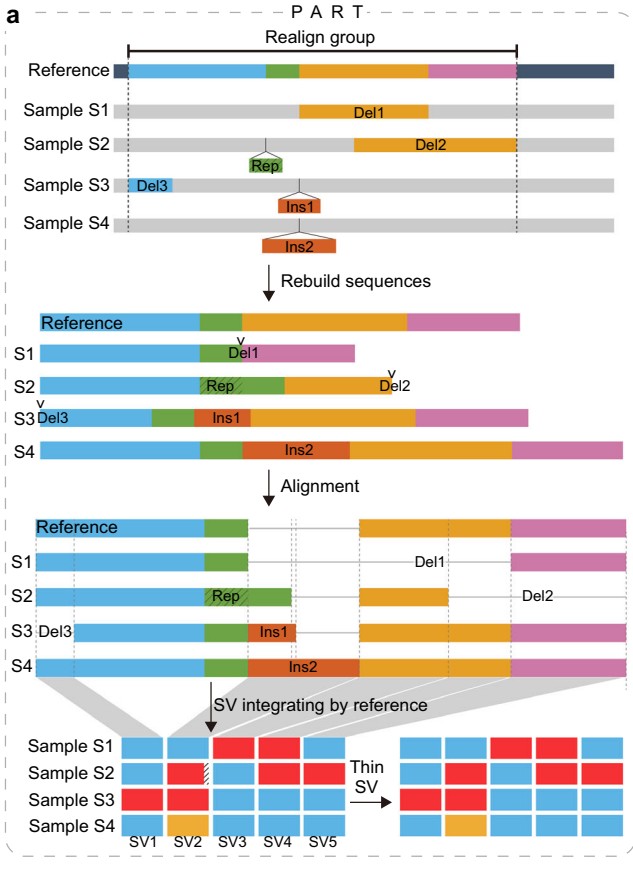

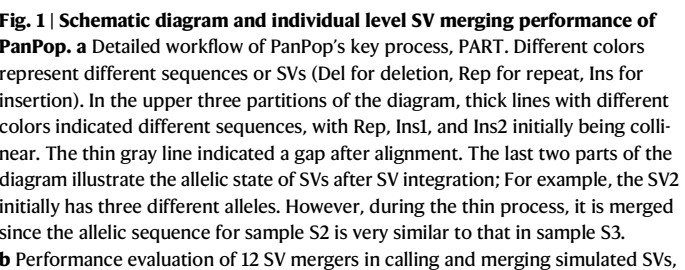

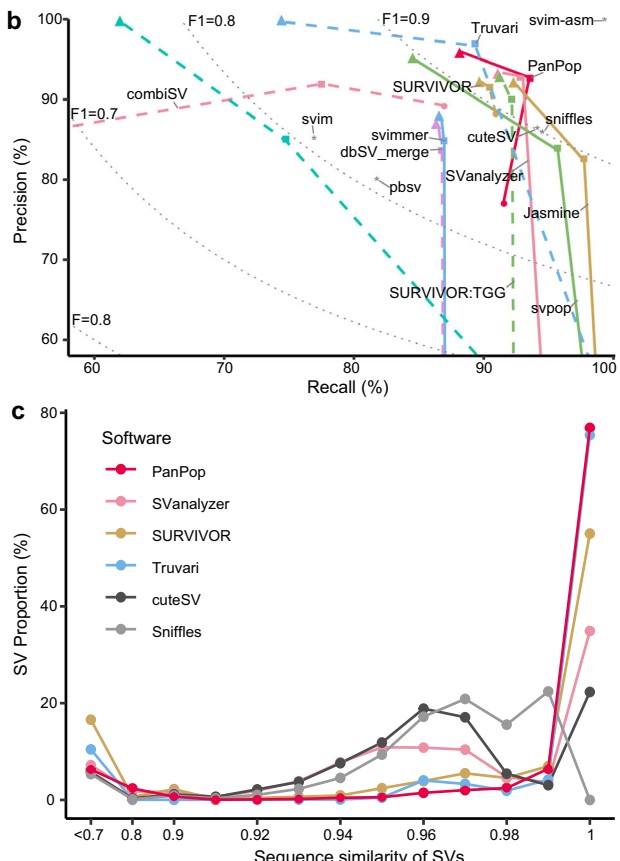

**Fig. 1 | Schematic diagram and individual level SV merging performance of PanPop. a** Detailed workflow of PanPop's key process, PART. Different colors represent different sequences or SVs (Del for deletion, Rep for repeat, Ins for insertion). In the upper three partitions of the diagram, thick lines with different colors indicated different sequences, with Rep, Ins1, and Ins2 initially being colinear. The thin gray line indicated a gap after alignment. The last two parts of the diagram illustrate the allelic state of SVs after SV integration; For example, the SV2 initially has three different alleles. However, during the thin process, it is merged since the allelic sequence for sample S2 is very similar to that in sample S3. **b** Performance evaluation of 12 SV mergers in calling and merging simulated SVs, assessed by recall (x-axis), precision (y-axis), and F1 score (F1, dashed line). Circles, squares, and triangles represent SVs supported by at least one (unfiltered), two, and three (filtered) SV callers for SV merges, respectively, while SV callers were marked as asterisks. SV mergers and callers are represented using different colors, line types, and shapes. **c** Distribution of sequence similarity among merged SVs from two SV callers and four SV mergers with F1-score higher than 0.9. Note that for SV mergers, only filtered SVs (SVs supported by at least two SV callers) were used and excluded SVIM-asm, which had an F1-score of 1. Source data are provided as a Source Data file.

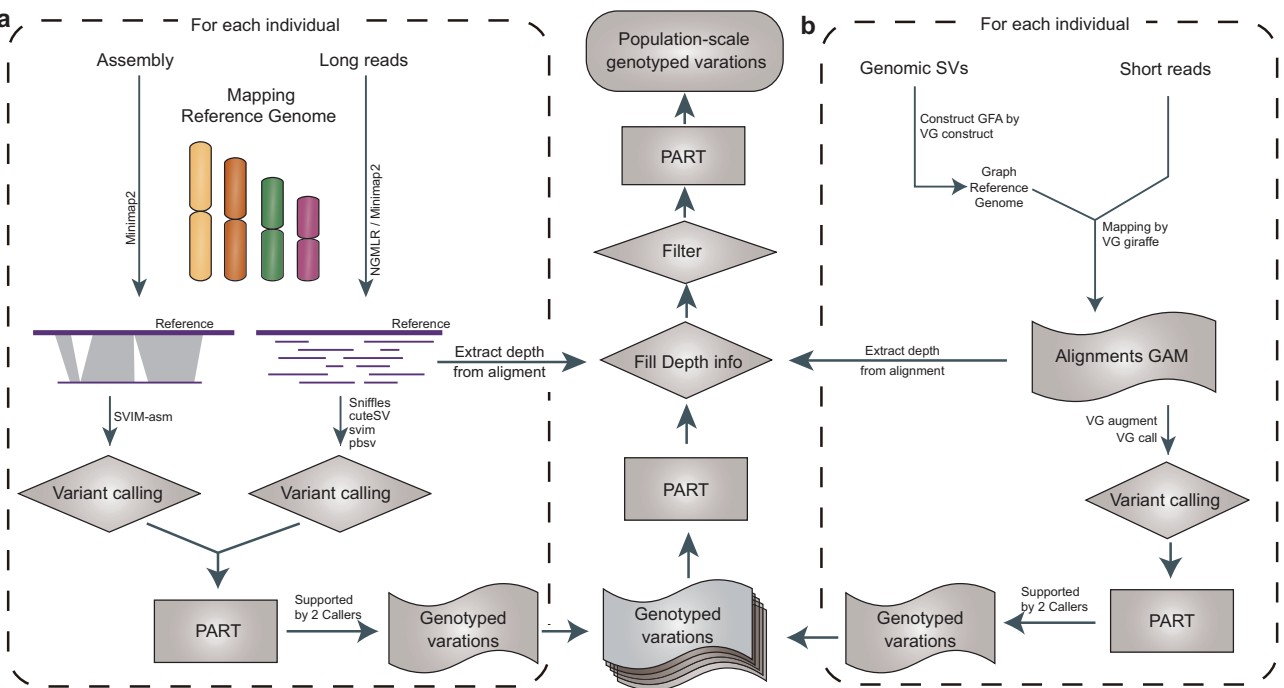

**Fig. 2 | Overview of the pipeline utilized by PanPop.** Pipelines for Long-reads-based and short-reads-based approaches were illustrated by **a**, **b**, respectively. The middle partition of the figure was shared by both pipelines.

called using SVIM-asm[25] (Fig. 2a). For short sequencing reads commonly used in population-level analyses, the VG toolkit[26] is employed for alignment to the graph-genome (pan-genome) reference and SV calling (Fig. 2b).

The PART method is applied twice to merge SVs from different callers, either for individual analysis or for multiple individuals within large populations. In the case of individual SV filtering, only SVs supported by two or more SV callers are retained when multiple callers are utilized. For the population-level SV filtering, referred as "fill-depth information", depth information is extracted from the mapping file for each SV in each sample and combined into the merged VCF file. During this process, SVs with abnormal depth are soft-masked to ensure result accuracy[15,27]. Additionally, although optional, it is recommended to filter SVs based on the maximum missing rate to obtain a final set of refined population-scale SVs. We also provide tools for aggressively thinning SVs by removing low-frequency alleles, resulting in a set of predominantly bi-allelic SVs.

## Evaluation of the accuracy of single individual among multiple approaches

To assess the performance of the PanPop pipeline, we conducted an evaluation that involved analyzing its ability to merge individual-level SVs using both simulated and real long-sequencing data. The software versions and detailed parameter used were available in Supplementary texts. Initially, we employed five SV callers to identify SVs, and then compared the results obtained using PanPop with those from nine other SV mergers. For the simulated *Arabidopsis thaliana* data, the F1-scores of the five SV callers ranged from 0.81 to 1.00, with variable recall and precision rates (Figs. 1b and 3d, Supplementary Data 1). The SVIM-asm achieved an almost perfect score in this simulation dataset. However, after merging without any filtering, the 12 mergers exhibited relatively low F1-scores, primarily due to low precision (Fig. 1b, Supplementary Figs. 1a and 2). Consequently, filtering was necessary. We tested the SVs that were supported by at least one or more callers and observed that the precision rate increased with the number of supporting callers, while the recall rate decreased significantly (Figs. 1, 3,

4). As a result, the strategy of requiring support from at least two SV callers yielded the best F1-score and we adopted this as the default parameter for subsequent analyses.

After applying filtering, both software programs demonstrated a significant increase in F1-score and precision rate, particularly with PanPop, Truvari and SVanalyzer achieving the highest F1-score, exceeding 0.93 (Fig. 1b, Supplementary Fig. 1a, Supplementary Fig. 2). To further assess accuracy, we calculated the similarity between the simulated SVs and the SVs identified by the 12 mergers and five SV callers (Figs. 1c and 3a). We found that PanPop identified a high proportion (83.3%) of high-similarity (≥98%). In comparison, the best two SV callers and three other mergers only showed proportions ranging from 25.4% to 79.6% of high-similarity SVs (excluding SVIM-asm, which had an F1-score of 1) (Fig. 1c). We also assessed the distribution of SV lengths, and found that PanPop exhibited a high degree of similarity (Fig. 3b, c). Collectively, these findings demonstrate that merging SVs can greatly improve accuracy, with PanPop delivering the best performance. Additionally, we evaluated the performance of PanPop using true data of high-confidence SVs in human HG002, and the results were mostly consistent with those from the simulated dataset (Fig. 4, Supplementary Fig. 1b). The F1-score of SVanalyzer was slightly higher than PanPop (0.958 vs 0.954), which may be attributed to the SV split strategy of PanPop, making benchmarking more challenging. The recall rate of SVIM-asm also decreased to 70% as the complexity of genomics increased. Moreover, the genotype accuracy of PanPop was significantly higher than that of SVanalyzer and Truvari, with values of 0.979, 0.463, and 0.920, respectively (Supplementary Data 2).

## Evaluation of the performance in the population-scale SV merging

We proceeded to assess the merging performance of population-scale SVs using two datasets of *A. thaliana*: 86 samples with long sequencing read and 1092 samples with short sequencing read. For population merging, it was necessary for the merger to generate genotype information for each individual to facilitate subsequent analyses. Therefore, out of the 12 mergers evaluated, only PanPop, SURVIVOR, bcftools,

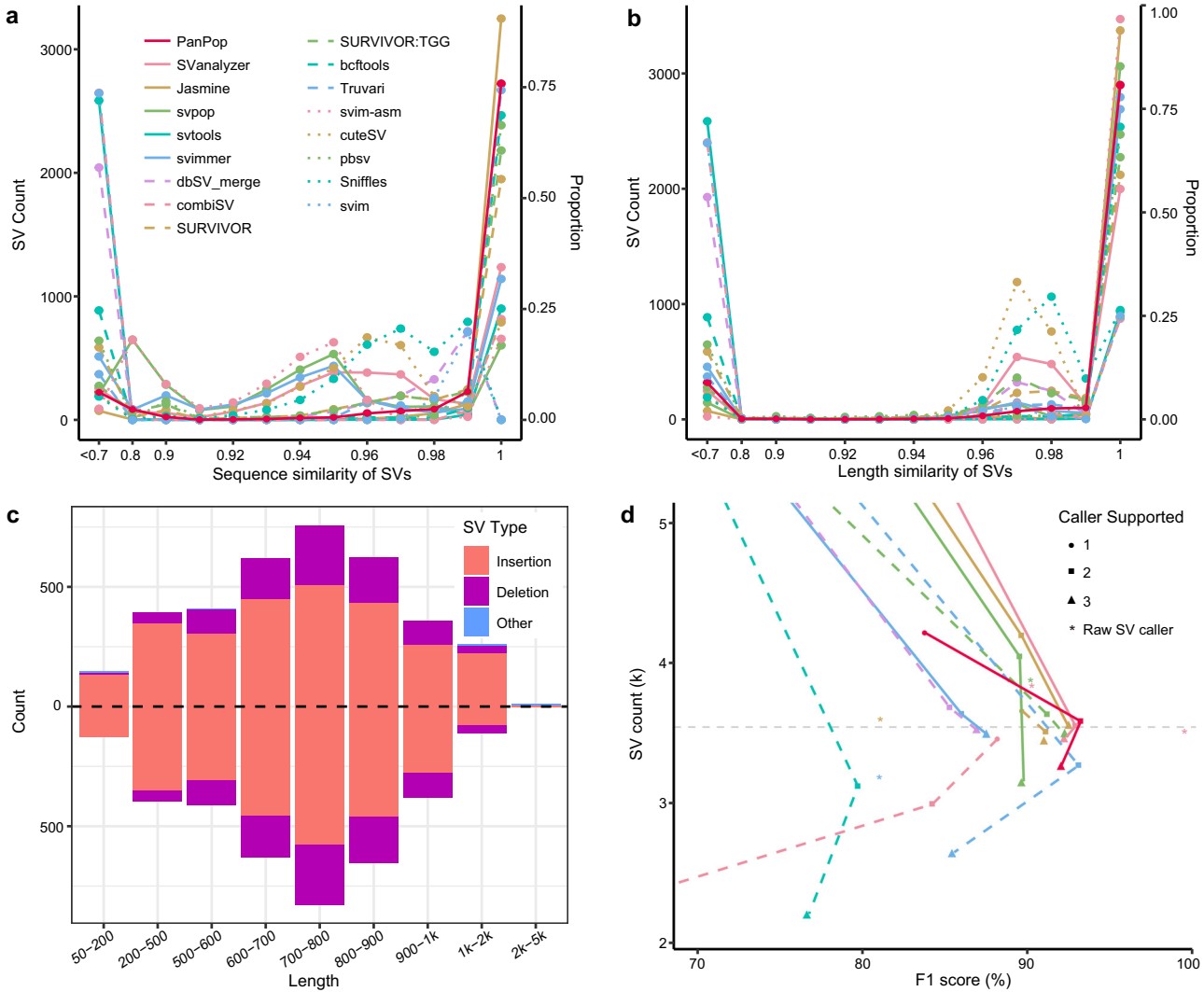

**Fig. 3 | Performance comparison of different SV mergers using a simulated SV dataset.** Sequence similarity **a**, length similarity **b**, and F1-score **d** comparison among 12 SV mergers with filtered SVs (SVs supported by at least two SV callers) and 5 SV callers. SV mergers and callers represented using different colors and line types. For **d** the gray dashed line was the simulated SVs count, and minimal supported SV caller count of 1 to 3 were represented as circle, square and triangle, respectively. **c** SV length distribution of simulated data (Lower) and PanPop result (Upper, filtered SVs). Red, ultramarine, and blue bars represent SV types of insertion, deletion, and other, respectively. Source data are provided as a Source Data file.

Truvari, and sv-merger were utilized. In the case of the long-sequenced dataset, SVs were identified by 5 SV callers and filtered by requiring support from at least 2 callers for each individual before merging all individual information into the population-scale. We initially evaluated the results based on the proportion of missing genotypes, as high missing rates are deemed meaningless in population analyses. The proportion of mutations with a missing rate exceeding 30% was 8.2% for PanPop, 62.2% for SURVIVOR, 78.1% for bcftools, and 99.7% for Truvari (Fig. 5a, Supplementary Fig. 3a) when merging 86 samples using the dataset filled with information of non-mutation samples by PanPop. When using the raw dataset generated by SV callers, the missing rate was exceedingly highly, exceeding 97% for each of these four SV mergers. The high missing rate of the raw dataset made it unsuitable for subsequent analyses, especially for bcftools and Truvari, primarily because their merging strategy involved nearly lossless bulk combinations of all mutations without processing (Figs. 5f and 6f). This distinction became more pronounced when merging 1092 samples (Fig. 6a). The lower missing rate of PanPop can be attributed to the PART algorithm and the "fill-depth information" process. However, when applying a data source after 'fill-depth information' process by PanPop, the missing rate of SURVIVOR and bcftools greatly dropped

but was still considerably higher than that of PanPop (Fig. 5a, Supplementary Fig. 3a). Alongside the missing rate, the proportion of biallelic SVs is also crucial, since multi-allelic SVs are challenging to utilize in subsequent population genetics analyses. We discovered that approximately 400k SVs (89.0%) were biallelic when merged by Pan-Pop, with only a slight change after filtering out sites with more than 30% missing data (Fig. 5f). This count and percentage were significantly higher than those of SURVIVOR and bcftools after filtering. This distinction was also more prominent when merging 1092 samples (Fig. 6f).

We proceeded to further benchmark SVs by comparing them before and after population merging. Since PanPop splits complex SVs into several smaller SVs, we reassembled those small SVs before evaluation. Regardless of the missing rate, bcftools achieved the highest F1-score (nearly 1) because it simply combined all SVs. PanPop also performed well, with an average F1-score of 97.5% and a recall rate of 99.0%. However, SURVIVOR only had a recall rate of 69.4% and an F1-score of 73.0% (Fig. 5d). After filtering SVs with a maximum missing rate of 30% (where a maximum of 30% of samples had missing genotypes for each SV), the performance of bcftools, Truvari and SURVI-VOR underwent significant changes. The F1-score of bcftools and

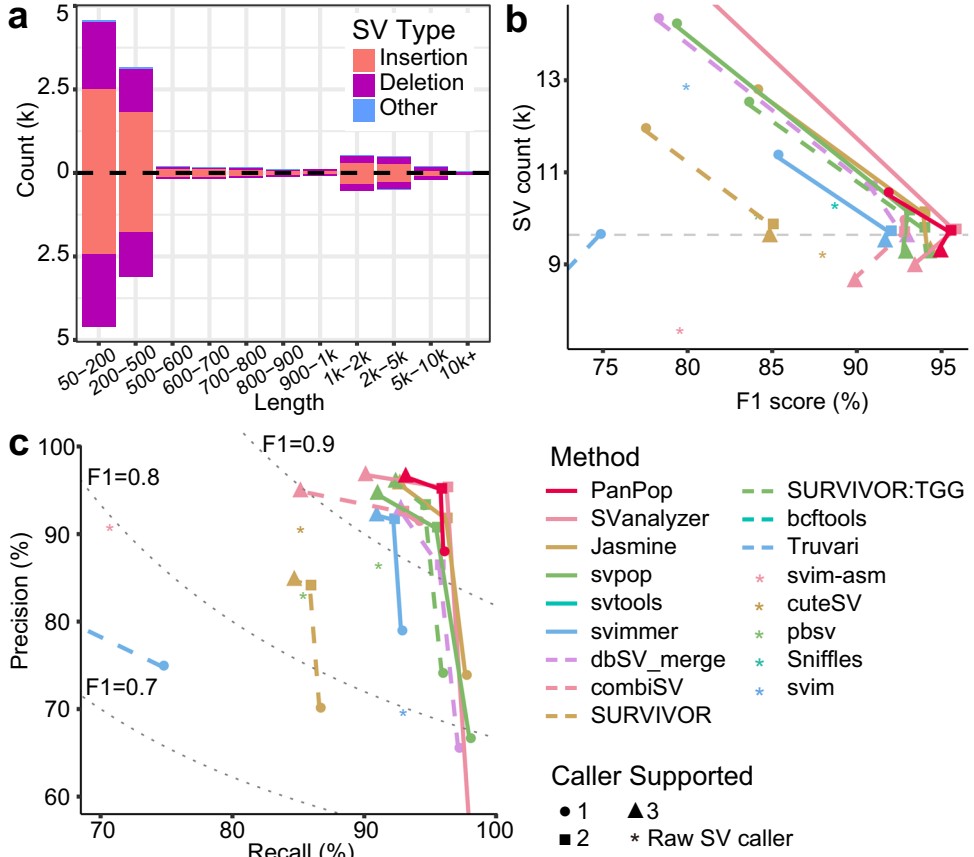

**Fig. 4 | Performance evaluation of PanPop using HG002 dataset. a** SV length and type for PanPop (upper, filtered SVs) and high-confidence dataset (lower). Red, ultramarine, and blue bars represent SV types of insertion, deletion, and other, respectively. **b**, **c** comparison between 12 SV mergers and 5 SV callers. For each merger, minimal supported SV caller count of 1 to 3 was represented as circle, square, and triangle, respectively. SV mergers and callers are represented using different colors and line types. The gray dashed line in b was the count of high-confident SVs, and the gray dashed line in c was F1 score of 0.9, 0.8, and 0.7. Source data are provided as a Source Data file.

Truvari nearly dropped to zero due to the high missing rate, while SURVIVOR's F1-score was greatly reduced to an average of 19.7%. However, due to its low missing rate, PanPop still achieved an F1-score of 95.4% (Fig. 5c). We also compared PanPop with the other two software programs capable of performing population-scale merging, namely sv-merger (Supplementary Note 1) and Sniffles2 (Supplementary Note 2), and found that their performance was not as good as PanPop (Supplementary Figs. 4 and 5). Considering the anticipated increase in sequencing data volume in the future, a large number of samples will be analyzed. Therefore, we used 1092 short-sequenced samples as a replacement to evaluate large-scale population SV merging. The VG toolkit was selected to align reads and call SVs, using the graph-genome generated by the 86 long-sequenced samples as the reference. The benchmark results of the four SV mergers were similar to those of the 86 samples, with PanPop demonstrating the best performance (Fig. 6, Supplementary Fig. 3b).

## Discussion

Despite the advancements made in detecting structural variations (SVs) using various SV callers and mergers, accurately combining SVs still poses a challenge[15]. This results in a fragmented SV analysis workflow and the possibility of introducing unnecessary errors[28]. In this study, we have developed the PanPop pipeline, which incorporates a sequence-aware merging algorithm named PART. This algorithm allows for the merging of SVs at both the individual and large population scales, while maintaining high-accuracy genotype information.

In addition to combining SVs from individual samples, we observed distinct behaviors among 12 SV merger tools and 5 SV callers. SVIM-asm achieved nearly perfect results for the simulated dataset of *A. thaliana* but its performance was mediocrity for the real HG002 dataset (Figs. 1b and 4c). This difference can be attributed to the simplified conditions present in the simulated dataset, which lacked SNPs and InDels and only contained homologous SVs. When confronted with more complex SV scenarios, assembly-based strategies like SVIM-asm exhibited a significant decrease in accuracy. Similarly, Truvari performs well on the simulated dataset but poorly on the real dataset (Figs. 1b, 4c). Considering its subpar performance in population-scale SV merging as well, we believe that Truvari is better suited for merging simple and highly consistent SVs and may struggle with partly overlapped SVs, similar to the performance of bcftools (Figs. 5d and 6d). On the other hand, SVanalyser showed excellent performance for both the simulated and real HG002 datasets (Figs. 1b and 4c), but it lacked the capability for population-scale merging.

When it comes to population-scale SV merging, the PART algorithm implemented in PanPop demonstrated significant advantages in terms of format/software compatibility and performance. Since most SV mergers struggle to merge SVs while preserving genotype information, we tested five SV mergers (PanPop, SURVIVOR, bcftools, Truvari and sv-merger) and along with Sniffles2, which is an SV caller that can only utilize its own individual results. The presence of extensive SV differentiation among samples posed a significant challenge for SV mergers, and both bcftools and Truvari failed to handle it (Figs. 5d and 6d). Instead of merging a large number of overlapping

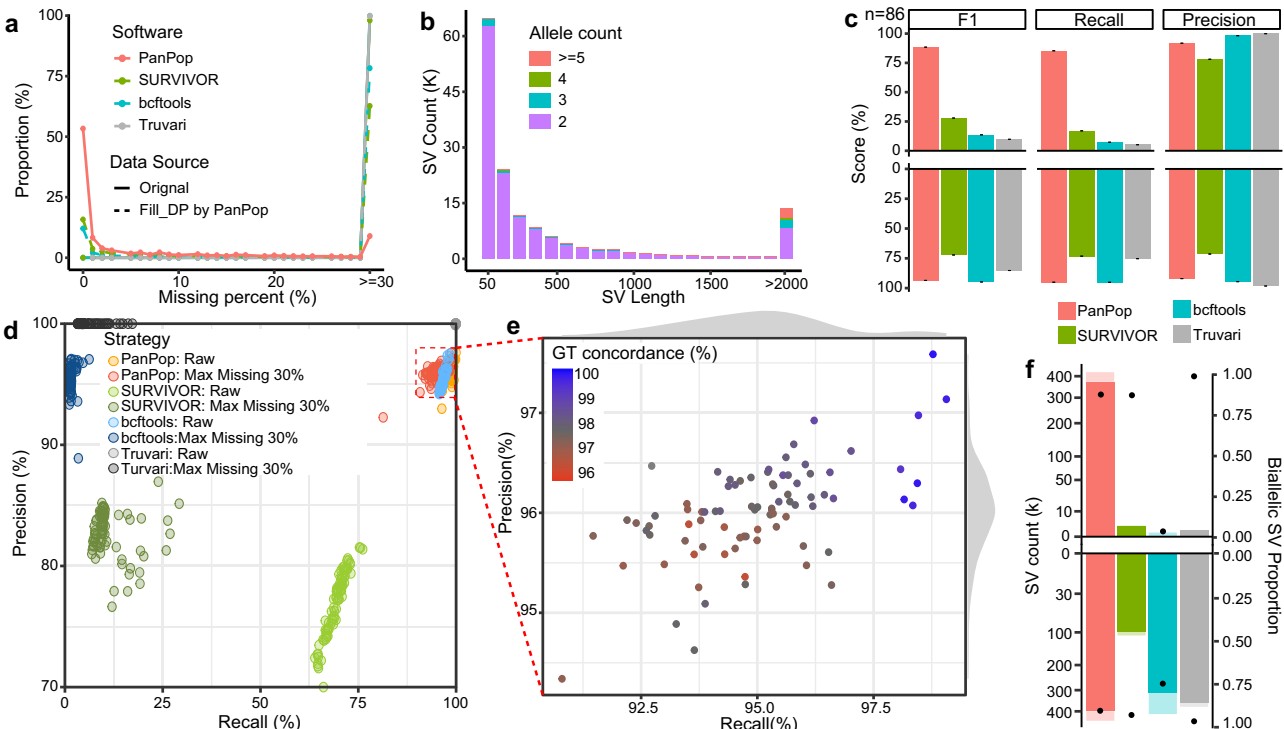

**Fig. 5 | Performance evaluation of the population-scale SV merging by four SV mergers using 86 *A. thaliana* long-sequencing datasets. a** shows the missing proportion of SVs. SV merging using the original results of SV callers or processed by PanPop's "fill_depth_information" method, represented by the solid or dashed lines, respectively. **b** SV length and allele count statistic for PanPop, with allele counts distinguished using different colors. **c, f** show the histogram of evaluation scores (recall, precision, and F1) and biallelic proportion of the results from four SV mergers. Bar and error bars of **c** are means ± SE ($n = 86$ for each bar). The upper and lower sections of **f** represent the datasets after a maximum missing rate filter of 30% and the raw datasets, respectively. The solid and shadow bars of **f** represented the count of biallelic and total SVs, respectively. The black dot of **f** represented the proportion of biallelic SV. **d** the precision and recall rate of four SV mergers. The bright and dark points represented the result of raw SVs and SVs with a filter of max missing of 30%, respectively. **e** Magnified view of the filtered PanPop results. For **e**, the color of each dot represents the genotype concordance (GT concordance), with the color gradient from red to blue indicating increasing numbers. For **a, c, d**, and **f**, colors of red, green, blue, and gray represented different software of PanPop, SURVIVOR, bcftools, and Truvari, respectively. Source data are provided as a Source Data file.

SVs, these programs stacked them, resulting in a high missing rate (Figs. 5f and 6f). In addition, our tests revealed limitations in the simple position-only merging algorithm employed by the widely used SUR-VIVOR, which resulted in the lowest precision rate among five SV merges (Figs. 5d and 6d). By incorporating a "Pre-clustering" algorithm, sv-merger achieved decent performance, closely resembling that of PanPop (Supplementary Fig. 4). However, sv-merger is unable to split large complex SVs into smaller, simpler SVs, and it requires complex transformations to be used effectively. Some SV callers, like Sniffles2, also support population-scale multi-sample SV calling. However, Sniffles2 introduced a specific format to store genotype information from each sample, making the merging process less universally applicable. Leveraging this advantage, Sniffles2 is the only SV merger that fills in the information of non-mutation samples, which is crucial for subsequent genomic analyses (Supplementary Fig. 5).

We acknowledge that PanPop requires more computational resources (CPU time and memory usage, Supplementary Table 2 and 3, Supplementary Note 3) compared to other software. This is mainly due to the sequence-aware alignment process, which improves accuracy and the fill-depth-information process while reducing the missing rate. However, PanPop was designed as a multithreading software capable of handling large volumes of data. With sufficient computational resources, PanPop can significantly reduce the overall elapsed time and provide the most accurate and applicable results. In summary, our developed PART algorithm demonstrates high-performance SV merge for any source of SVs, regardless of the number of SV callers used, whether in a single individual or a population with sample sizes ranging from hundreds to thousands. The entire PanPop pipeline encompasses the complete process from reads to SVs, making SV calling and merging accessible to anyone for SV-related analyses without the need for specific thresholds.

# Methods

## Sequencing data
Accurate assessment of SV callers and mergers for individual analysis requires a high-confidence dataset. For this purpose, the HG002 cell line has been extensively utilized, and a high-confidence SV call set has been generated through consensus calling using multiple sequencing technologies. Additionally, the Genome in a Bottle (GIAB) project provides a trust set that includes high-confidence regions of the HG002 dataset (HG002_SVs_Tier1_v0.6.-bed), encompassing 9641 SVs. In our evaluation, we utilized publicly available HiFi reads (PacBio CCS 15 kb dataset) with an average sequencing depth of 28×. Additionally, we employed VISOR[29] to simulate SVs and generate corresponding HiFi reads of *A. thaliana* (see Supplementary Note 4).

For population-scale assessment, we selected A. thaliana as a model plant due to its abundant sequencing data and relatively small genome size. The most recent T2T genome, Col-PEK, served as the reference, and we analyzed 86 third-generation sequencing (TGS) samples (refer to Supplementary Data 3). We excluded Col samples themselves and samples with SVs fewer than 100, resulting in a total of 81 HiFi and 5 ONT sequenced samples. All the data used in this study were publicly available from NCBI or CNCB databases. The average

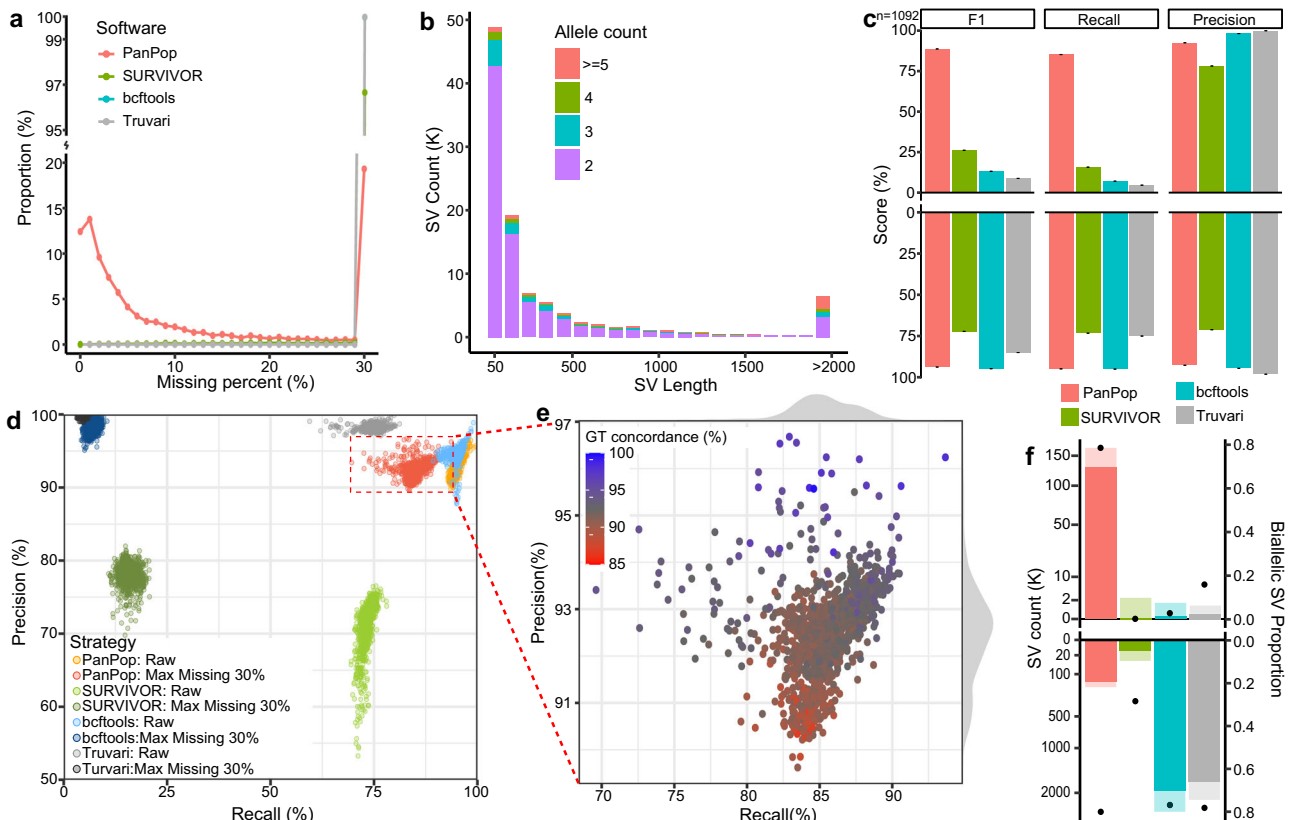

**Fig. 6 | Performance of four SV merger in large-scale SV merging using 1092 *A. thaliana* NGS datasets. a** shows missing proportion of SVs. SV merging using the original results of SV callers. Line of bcftools and Truvari were mostly overlapped. **b** SV length and allele count statistic for PanPop, with allele counts distinguished using different colors. **c, f** shows the histogram of evaluation scores (recall, precision, and F1) and biallelic proportion of the results from four SV mergers. Bar and error bars of **c** are means ± SE (*n* = 1092 for each bar). The upper and lower sections of f represent the datasets after a maximum missing rate filter of 30% and the raw datasets, respectively. The solid and shadow bars of **f** represented the count of biallelic and total SVs, respectively. The black dot of **f** represented the proportion of biallelic SV. **d** the precision and recall rate of four SV merger. The bright and dark points represented the result of raw SVs and SVs with a filter of max missing of 30%, respectively. **e** Magnified view of the filtered PanPop results. For **e**, the color of each dot represents the genotype concordance (GT concordance), with the color gradient from red to blue indicating increasing numbers. For **a, c, d** and **f**, the color of red, green, blue, and gray represented different software of PanPop, SURVIVOR, bcftools, and Truvari, respectively. Source data are provided as a Source Data file.

depth of the 86 samples was 66.0×, with depths ranging from 16.1× to 140.6×.

To evaluate SV analysis in a large population context, we obtained Illumina short-read data for 1135 samples from the *Arabidopsis* 1001 Genomes Project[28]. After excluding 42 samples due to short read length and Col-0 sample itself, we retained 1092 samples with an average sequencing depth of 31.9 (Supplementary Data 4).

## PanPop's PART method

We used our 'realign' algorithm to ensure accuracy during the splitting or combining of mutations based on sequences. Firstly, we divided SVs into realign groups based on their positions. Each realign group had specific base pairs (e.g., 200 bp) gap between them, and within each realign group, the gaps were limited to a certain base pair threshold (also 200 bp in this case) (Fig. 1a). Within each realign group, we reconstructed the consistent sequence of each individual by replacing REF bases with ALT sequence. To prevent sequence overlapping, we manually reversed the second phase of the second hetero mutations. PanPop also tried to resolve SV overlapping by removing the overlapping sequences, which ideally should not happen and may be attributed to incomplete SV callers. Then the alignment was processed for these haplotypes, and the aligned sequences were anchored to the reference sequence to determine and split each SVs (Fig. 1a). Because the range of realign group extends a certain base pair distance (in this

case, 200 bp), duplicated SVs or SVs in repeat region can be aligned and ordered correctly. Given that the realign group could be quite long, we employed multiple software options for global alignment. We used MUSCLE for short realign groups and FAMSA and stmsa for long realign groups respectively. No significant differences in accuracy were found between the repeat and non-repeat regions (Supplementary Table 4 and Supplementary Note 5).

For multi-allelic mutations, we performed a pairwise comparison of each pair of alleles and merged them if they showed similarity. We aligned each pair and calculated the differences in bases and gap lengths. If at least 60% of the bases in two sequences were identical and the count of different bases was below 20, we considered these two sequences to represent the same allele (Supplementary Note 6, Supplementary Fig. 6). After comparing each pair, we employed the MCL algorithm to identify groups of alleles that were deemed the same. Within each MCL group, the alleles were merged.

In many instances, filtering was implemented following the thinning process, which also aimed to streamline SVs. As numerous SVs occurred at low frequencies, they presented difficulties for subsequent analyses or enrichment. Therefore, we recommended discarding alleles with a frequency of less than 1% after the thinning process. It is important to note that the PART method was developed to handle various sources of SVs, not limited to the PanPop pipeline. Consequently, a standalone version of PART was also made available.

## Variant calling for TGS

To detect genome-wide structural variations in the query de novo assemblies, we utilized the SVIM-asm tool as the caller for the assembly-based strategy. Each query genome was aligned to a reference genome using the minimap2 software with the following parameters: "-a -x asm5 --cs -r2k". Subsequently, SV identification was carried out by the SVIM-asm software using the 'haploid' command and default parameters.

To ensure accurate SV detection, we made use of raw sequencing reads, particularly TGS (Third Generation Sequencing) data. ONT (Oxford Nanopore Technologies), PacBio, or HiFi reads were aligned to the reference genome using NGMLR. It is important to mention that we included the parameter "-x ont" for aligning ONT data, as per the recommended guidelines. Subsequently, we employed four structural variant callers for the read-based strategy: Sniffles (v2.0.7), SVIM (v1.4.2), cuteSV (v2.0.2), and pbsv (v2.9.0), all with default parameters (see Supplementary Table 5 and Supplementary Note 7 for details).

To consolidate the mutations identified by multiple callers, we utilized the "merge" tool provided by bcftools with the parameters "-m none". The merged mutations were then subjected to further refinement and realignment using PanPop, resulting in a raw set of population mutations. However, the raw reads still contained several missing sites. To determine whether these sites were in the reference state or unknown (due to mis-sequencing or misalignment), we extracted depth information for each site in each sample from the alignment files and filled them in the population mutation set accordingly. We applied additional filters to softly mask sites with excessively low depth (less than 1/4 of the individual average depth) or excessively high depth (more than 4-fold the individual average depth). Lastly, the population mutations were realigned and pruned to generate the final set of population mutations in VCF (Variant Call Format) format. A parser was applied to reconcile output of different SV callers (Supplementary Note 8).

## Structure variant accuracy and recall assessment

The assessment of different SV callers and mergers was conducted using the 'bench' module in Truvari software (v3.0.0)[29], with the following parameters: 'bench -p 50 -P 50 -r 500'. To evaluate the performance of SV callers, we compared their results with a dataset of high-confidence SVs. Regarding the merging process, we compared SVs before and after population merging for each individual. However, since PanPop has the capability to split large and complex SVs into smaller and simpler ones, it is crucial to merge the small SVs for each individual after extracting them from the population SV sets. To facilitate this, we generated a list of ranges for each realigned SV before population merging. After extracting the SVs for a single individual from the population SV sets, only the SVs located within that list were merged. This approach helps to reduce the likelihood of over-merging or under-merging. To be fair, we also added a benchmark of SVs with the base SVs not been realigned and available at Figshare [https://doi.org/10.6084/m9.figshare.25021043], which shows a same trend of the realigned base SVs. This approach helps minimize the risk of over-merging or under-merging. It is important to note that for the assessment of HG002, only SVs located within high-confidence regions (HG002_SVs_Tier1_v0.6.bed) were utilized. During subsequent population-scale analysis, the presence of missing data can significantly impact many results[16,17]. Therefore, we performed an additional test for each SV merger, filtering out sites with a missing rate greater than 0.3. This process was also applied to other competition SV merging software and version with detailed parameters available in Supplementary Table 1 and Supplementary Note 7.

## Structure variant for NGS data

The PanPop pipeline was employed in 'NGS-Augment' mode to process the graph-genome and raw NGS reads for 1092 A. thaliana individuals. The process is outlined below. First, a graph-genome was constructed based on all mutations obtained from 86 long sequences of A. thaliana using the construct module of the VG toolkit. Second, for each of the 1092 individuals, the cleaned reads were mapped to the pangenome using the giraffe function within the VG toolkit (v1.36.0)[26], with default parameter settings. To generate a new type of structural variations (SVs) similar to those detected using long-reads, the augment mode in the VG toolkit was utilized. Third, augmented reads and the graph-genome were then evaluated for read support using the 'pack' function and genotyped using the 'call' function. Finally, the merging of mutations for individuals to generate population-scale mutations followed a similar process as in TGS (Third Generation Sequencing) data. However, since a graph-genome was employed, the depth of each sample was calculated using the 'depth' module of the VG toolkit.

## Reporting summary

Further information on research design is available in the Nature Portfolio Reporting Summary linked to this article.

## Data availability

We used publicly available sequencing data in this study. The HiFi sequencing data for HG002 are available at GIAB FTP site [https://ftp.ncbi.nlm.nih.gov/ReferenceSamples/giab/data/AshkenazimTrio/HG002_NA24385_son/PacBio_SequelII_CCS_11kb]. Raw NGS datasets of *Arabidopsis thaliana* are available under BioProject accession PRJNA273563. Raw TGS datasets of *A. thaliana* are available under BioProject PRJCA012695, PRJEB55353, PRJNA715329 and PRJNA834751. The dataset of single individual SV merging in the *A. thaliana* simulated dataset can be accessed at Figshare. The HG002 dataset is also available at Figshare. Population-scale SV merging results and raw SV data for each TGS sample can be found in Figshare. Results of 'Truvari bench' comparison between PanPop and other software can be found in Figshare. Source data are provided in this paper.

## Code availability

PanPop package is available at GitHub [https://github.com/starskyzheng/panpop] under MIT license. Code for reproducing and detailed parameters are available in Supplementary Note 7. An example of PanPop and PART were available from CodeOcean [https://doi.org/10.24433/CO.1577027.v1].

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

## Acknowledgements

Financial support was provided by the National Key R&D Program of China (2023YFF0805600 to Y.Y.), the Key Science & Technology Project of Gansu Province (22ZD6NA007 to Y.Y.), the National Natural Science Foundation of China (32170219 to Y.Y.), the Fundamental Research Funds for the Central Universities (lzujbky-2022-ey07 to Y.Y.) and the Young Talent Development Project of State Key Laboratory of Herbage Improvement and Grassland Agroecosystems (No. 2021 + 02 to Y.Y.), and International Collaboration 111 Program (BP0719040). All the computation works were supported by three platforms: Supercomputing Center of Lanzhou University, Lanzhou University Big Data and Intelligence Experimental Platform for Performance Governance and Public Safety: Performance Governance and Management Big Data Laboratory, and Big Data Computing Platform for Western Ecological Environment and Regional Development.

## Author contributions

Y.Y. and J.L. conceived this project. Z.Z. devolved the PanPop software and carried out most of the analysis. M.Z. and J.Z. collect datasets. X.L., L.H. and W.L. participated in data testing and graphing tasks. S.Y., C.L. and X.Y. helped with computing analyses in the revision stage. Z.Z., Y.Y. and J.L. wrote the paper. All of the authors read and approved the final manuscript.

## Competing interests

The authors declare no competing interests.
