## [Peer Review File · Nature Communications]

PanPop: A Sequence-aware Merger of Genomic Structural Variations at Population scaleReviewers' Comments:

Reviewer #1:

Remarks to the Author:

In this study, the authors introduced a tool named PanPop that allows to call and merge SV variants of various types, thus providing a population-level SV set. They tested it using simulated and real datasets. They also benchmarked SVs obtained with their tool and other available software solutions for merging SVs.

The problem set by the authors, namely the accurate identification of SVs in both individual genomes and entire populations - and especially reducing them to biallelic SVs - which would make them more applicable e.g. to GWAS or other comparative studies - is of high importance. Thus, in my opinion, it will make a valuable contribution to the field. As a non-bioinformatician, I could only assess the work from the point of view of a potential end user of the software outputs. The idea of SV merging is overall clearly presented. Splitting and realigning more complex SVs based on position seems to be the good trick to finally resolve them and simplify to biallelic variants. For SV calling module the authors used well-recognized detection tools. The presented benchmarking results indicate that PanPop nicely does its work. The tool could be installed and run on example data without major issues. Overall it looks very promising, but since the manuscript is a general description of the software, not its application to a specific research project, my conclusions are limited.

Comments:

Maybe the authors could consider including the actual results (merged SVs for Arabidopsis) or provide some representative examples of SV regions that could be well resolved this way? This could actually help to evaluate PanPop performance in the background of existing knowledge on Arabidopsis variation, especially that the authors used datasets previously utilized in SV analysis projects (e.g. 10.1016/j.cell.2016.05.063, 10.1038/s41467-023-42029-4, <https://doi.org/10.1105/tpc.19.00640>, <https://doi.org/10.1093/nar/gkab904>)

It is not clear to me whether and how this merging step (described in the manuscript as a PART method) could be treated as an independent module. Can it be used with SVs originating from other sources than the SV detection tools integrated in PanPop? If so, it could be better explained in the text and the documentation, how to run the merging step independently from the SV detection.

Minor comment: Table S2 – why are the first seven accessions in the table marked as Duplicated of Col-PEK and excluded, while only one of them (first) is actually Col-0, according to the information provided in this table? On the other hand, Col-0 (id 6909) has been NOT excluded from the short read dataset (Table S3), although I do not expect to make it a real difference in this case, except that probably few SVs should be found in this particular sample.

Reviewer #2:

Remarks to the Author:

This manuscript describes a method called PanPop, designed for merging and genotyping structural variants (SVs). This problem becomes increasingly important as more high-quality long-read based SV calls become available. The key difference from the previous algorithms is that PanPop is using multiple sequence alignment, which allows it to better reconcile multiple allele variants without reference bias. On simulated data PanPop compares favorably against the current methods, although the improvement over SVanalyzer seems fairly small. On merging real population-scale SV calls, PanPop had the most biallelic variants and fewer uncertain genotype calls. I have a few questions below.

1. I believe that truvari can also perform SV merging using the collapse mode. Since it's currently one of the most popular methods for SV evaluation and merging, it needs to be included in the comparison.

2. In the second part of their evaluation (focused on the population merging), the authors excluded

most of the tools because they do not generate genotype calls. However, PanPop, in fact, performs two tasks here: merging and genotyping. Most methods from the authors multi-tool comparison were specifically designed for merging, while other genotyping tools also exist (e.g. Sniffles). It would be fair to include a few best performers of the multi-tool benchmark to the population-scale benchmark.

3. In addition to Figures 1 and 2, the numerical values for recall/precision/F1 scores for all tools should be given, perhaps as a supplementary table. It would also be informative to include the total number of SV calls and counts of TP/FP/FN for each tool.

4. Multiple sequence alignment methods are typically more computationally intense. How does running time and memory usage of PanPop compare to the other tools?

5. The authors used custom parameters for truvari, in particular `-mutimatch`? What was the intuition behind that? It would be informative to see the benchmarks using default parameters on a representative dataset (as a supplementary table).

6. When building a realign group, what happens to SVs that are within the clustering distance, but on different haplotypes? What happens in the case of overlapping SVs?

7. The authors use the term "maximum missing rate" to benchmark the methods on a population dataset. It needs to be clearly defined in the text. Is this a proportion of samples with missing genotypes?

8. In these benchmarks, Assemblytics seems to be performing poorly. This is a fairly old method, and more recent assembly-based SV callers, such as SVIM-asm or dipdiff seem to have better performance. It might make sense to include one of them into the comparison.

9. Line 169: is the dataset Nanopore or PacBio?

10. Results from all tools that were used for evaluation should be made available via a public repository.

11. In supplementary information, the authors state that the software is free for academic, but not commercial use, but the license on github repository seems to be MIT.

Reviewer #3:

Remarks to the Author:

In this study, the authors described a new method for merging structural variations, which was evaluated in detail using simulated dataset, individual dataset, and population-scale dataset. The study addressed an important issue in the field. The authors demonstrated that the method had commendable performance with potential applications. However, there are some concerns that require clarification.

Major concerns:

1. The experiments mainly evaluated the performance of PanPop on simulated and Arabidopsis thaliana datasets. It is suggested to further validate PanPop on real data from other model animals, in order to further demonstrate that PanPop has good adaptability to different species.
2. Unlike the non-repeat region, the breakpoint coordinates of SV obtained by different SV callers are inconsistent in the tandem repeat region. Inconsistent breakpoints affect SV merging. The authors need to take this into account in both simulated and real datasets. They may take different consideration for different regions.
3. Line 51, MUSCLE is acknowledged as a reliable multiple sequence aligner. However, when dealing

with numerous repeated sequences and significant variations in sequence length, misalignments may occur in the comparison results. How does the alignment accuracy affect the determination of biallelic and multiallelic SVs?

4. Give more details on the parameter settings and principles behind key parameters chosen for the PART algorithm. For example, lines 61-62 mention a threshold for the group length, but the rationale behind this choice is not clear. Is there a specific reason for selecting this threshold? If the group length threshold is modified, should the 20 bp difference also be adjusted?
5. How are large SVs handled, such as deletions larger than one or two realignment groups? If a large SV spans multiple groups, is there a risk of it being split into multiple biallelic SVs?
6. Please provide clear instructions on how to obtain the sequence during the "rebuild sequences" step in the pipeline. Is this step based on the BAM file or the VCF file? Not all VCF files from SV calling tools contain sequence information in the Info column.
7. Please evaluate the memory consumption and speed of the software.

Minor concerns:

1. The author may compare PanPop to sv-merger from the Icelandic population SV article (<https://doi.org/10.1038/s41588-021-00865-4>). (<https://github.com/DecodeGenetics/sv-merger>)
2. Please expand the description of existing SV merging tools. Highlight their limitations in handling complex overlapping SVs.
3. Please sharpen the conclusions by summarizing the key innovations over previous methods.
4. How is the simulated SV dataset constructed, described in Methods.
5. No other SV merger usage parameters, versions, and references are seen in Methods. Such as svtools, svimmer, SV-Pop, jasmine and other software.
5. In Figure 1b, both cuteSV and Sniffles exhibit higher F1-scores. Given that version 2 of Sniffles is capable of calling SVs at the population level, could you provide insights into the performance of Sniffles calls at the population level?
6. In Figures 1 and 4, the accuracy of individual SV callers appears generally low. To enhance clarity, it is recommended to present the results of two or three selected callers among the five.
7. Lines 163-170 refer to the 9641 SVs of HG002, but not all of them are SV sets. Could you elaborate on how precision and sensitivity are evaluated based on this dataset?
8. On line 176, it would be helpful to clearly specify the type of data.
9. There are some errors in the manuscript, such as the F1-score value in Figure 4.
10. Line 296: Updated reference to Jasmine, (Kirsche, M., Prabhu, G., Sherman, R. et al. Jasmine and Iris: population-scale structural variant comparison and analysis. Nat Methods 20, 408–417 (2023). <https://doi.org/10.1038/s41592-022-01753-3>)

REVIEWER COMMENTS

Reviewer #1 (Remarks to the Author):

In this study, the authors introduced a tool named PanPop that allows to call and merge SV variants of various types, thus providing a population-level SV set. They tested it using simulated and real datasets. They also benchmarked SVs obtained with their tool and other available software solutions for merging SVs.

The problem set by the authors, namely the accurate identification of SVs in both individual genomes and entire populations - and especially reducing them to biallelic SVs - which would make them more applicable e.g. to GWAS or other comparative studies - is of high importance. Thus, in my opinion, it will make a valuable contribution to the field. As a non-bioinformatician, I could only assess the work from the point of view of a potential end user of the software outputs. The idea of SV merging is overall clearly presented. Splitting and realigning more complex SVs based on position seems to be the good trick to finally resolve them and simplify to biallelic variants. For SV calling module the authors used well-recognized detection tools. The presented benchmarking results indicate that PanPop nicely does its work. The tool could be installed and run on example data without major issues. Overall it looks very promising, but since the manuscript is a general description of the software, not its application to a specific research project, my conclusions are limited.

Reply: We appreciate your positive and valuable comments.

Comments:

Maybe the authors could consider including the actual results (merged SVs for Arabidopsis) or provide some representative examples of SV regions that could be well resolved this way? This could actually help to evaluate PanPop performance in the background of existing knowledge on Arabidopsis variation, especially that the authors used datasets previously utilized in SV analysis projects (e.g. [10.1016/j.cell.2016.05.063](https://doi.org/10.1016/j.cell.2016.05.063), [10.1038/s41467-023-42029-4](https://doi.org/10.1038/s41467-023-42029-4), <https://doi.org/10.1105/tpc.19.00640>, <https://doi.org/10.1093/nar/gkab904>)

Reply: Good suggestion! We have followed your advice and uploaded all the raw results and intermediate files to figshare. This includes the data from various SV callers and all the SV mergers. The simulated dataset and VCFs of SV mergers for single individual SV merging are available at the following DOIs: [10.6084/m9.figshare.24657801](https://doi.org/10.6084/m9.figshare.24657801) for the *A. thaliana* simulated dataset and [10.6084/m9.figshare.24658671](https://doi.org/10.6084/m9.figshare.24658671) for the HG002 dataset. Additionally, the population-scale SV merge results and raw SV data for each TGS sample can be accessed at DOI [10.6084/m9.figshare.24659160](https://doi.org/10.6084/m9.figshare.24659160). By providing access to these datasets on figshare, we ensure transparency and enable other researchers to replicate and verify our findings. This part of information were added to data availability of our advised manuscript.

It is not clear to me whether and how this merging step (described in the manuscript as a PART method) could be treated as an independent module. Can it be used with SVs originating from other sources than the SV detection tools integrated in PanPop? If so, it could be better explained in the text and the documentation, how to run the merging step independently from the SV detection.

Reply: Actually, PART module is designed to handle SVs in VCF format, regardless of whether they were generated by PanPop or other SV detection tools. In short the PART module consists of two processes: realign and thin. The realign process involves joining and splitting nearby SVs to create several simpler SVs, while the thin process combines similar alleles to further simplify the SVs. It is important to note that the PART module is an independent module within PanPop and can be used separately from the rest of the pipeline. We have made revisions to the manuscript to include this information, specifically in lines 298-299. Additionally, we have added an entry point for the PART module at 'bin/PART_run.pl'. To provide users with a clear understanding of its usage, we have included detailed instructions in the supplementary text and an example on the PanPop software website on GitHub (<https://github.com/starskyzheng/panpop>).

Minor comment: Table S2 – why are the first seven accessions in the table marked as Duplicated of Col-PEK and excluded, while only one of them (first) is actually Col-0, according to the information provided in this table? On the other hand, Col-0 (id 6909) has been NOT excluded from the short read dataset (Table S3), although I do not expect to make it a real difference in this case, except that probably few SVs should be found in this particular sample.

Reply: We apologize for this confusion caused and appreciate your valuable comment. In our software evaluation process, we obtained the *A. thaliana* long reads dataset from the author of a paper published in BioRxiv (<https://doi.org/10.1101/2022.12.18.520013>), which has now been published in Nature Communications (<https://doi.org/10.1038/s41467-023-42029-4>). We used their dataset, which deleted the seven accessions. However, we did not consider their actual conditions in the previous version of our evaluation.

In order to comprehensively incorporate the SV dataset and ensure the inclusion of as many SVs as possible, we have conducted an evaluation of all the long-read samples to determine whether any samples need to be excluded from our dataset. We found five of the seven accessions and one other sample only contained very few SVs (range from 27 to 70), and in contrast, the others contained an SV number from 233 to 11,489. So, we deleted the six samples with SVs less 100 in the revised manuscript to construct the *Arabidopsis* SV dataset. Additionally, for the short reads analyses, we have excluded the Col-0 sample (id 6909). Finally, we have updated all the related information and corresponding results in this revised manuscript (Line 268-269, Tables S2, S3).

Reviewer #2 (Remarks to the Author):

This manuscript describes a method called PanPop, designed for merging and genotyping structural variants (SVs). This problem becomes increasingly important as more high-quality long-read based SV calls become available. The key difference from the previous algorithms is that PanPop is using multiple sequence alignment, which allows it to better reconcile multiple allele variants without reference bias. On simulated data PanPop compares favorably against the current methods, although the improvement over SVanalyzer seems fairly small. On merging real population-scale SV calls, PanPop had the most biallelic variants and fewer uncertain genotype calls. I have a few questions below.

Reply: Thank you for acknowledging the importance of our method and for providing valuable suggestions to help us enhance the data evaluation process.

1. I believe that Truvari can also perform SV merging using the collapse mode. Since it's currently one of the most popular methods for SV evaluation and merging, it needs to be included in the comparison.

Reply: Good suggestion. We have included Truvari in our comparison of software tools for SV merging in the revised manuscript. Truvari utilizes the result of `bcftools merge` and its performance also like bcftools, which cannot merge partly overlapped of complex mutations from different individuals. As a result, Truvari were able to process SVs of single individual but faced challenges when merging population SV. The corresponding results were added to the revised manuscript (Line 35, 120 151, 207, et al. Figs. 1, 3-6, supplementary figures S1, S3).

2. In the second part of their evaluation (focused on the population merging), the authors excluded most of the tools because they do not generate genotype calls. However, PanPop, in fact, performs two tasks here: merging and genotyping. Most methods from the authors multi-tool comparison were specifically designed for merging, while other genotyping tools also exist (e.g. Sniffles). It would be fair to include a few best performers of the multi-tool benchmark to the population-scale benchmark.

Reply: Thank you for your valuable comments, which will contribute to enhancing the fairness and validity of our comparison. It is indeed true that while there are tools available for SV merging in genotyping, they often have specific requirements regarding input and output formats. For example, Sniffles2's SV merging step is limited to utilizing only the SV dataset generated by itself. In our evaluation, we thoroughly assessed the performance of Sniffles2 at the population level and discovered that it exhibited superior performance in terms of recall and precision rates compared to the majority of other software tools. This finding underscores the strengths of Sniffles2 in delivering accurate results. However, it is important to emphasize that even with Sniffles2's improved performance, PanPop remains the highest-performing tool according to our comprehensive assessment, as depicted in both Figure S5 and Figure 5 of our advised manuscript (Line 86, 183, 217 and 231, et al.).

3. In addition to Figures 1 and 2, the numerical values for recall/precision/F1 scores for all tools should be given, perhaps as a supplementary table. It would also be informative to include the total number of SV calls and counts of TP/FP/FN for each tool.

Reply: We have added the detail results of each tool in different comparison at Table S5, S6, extended data, including the SV number, TP, FP, FN, recall, precision, F1 score and other related information.

4. Multiple sequence alignment methods are typically more computationally intense. How does running time and memory usage of PanPop compare to the other tools?

Reply: It is true that PanPop consumes more computational resources due to its inclusion of the MSA process, which results in higher precision. However, it is also worth noting that amongst the 12 software tools we tested, only PanPop and svpop can effectively use multithreading. Therefore, if additional computational resources are made available, PanPop could significantly reduce the elapsed time. We conducted a comprehensive evaluation of CPU time, memory usage, and elapsed time for both individual and population-scaled SV merging.

For single individual SV merging, PanPop had the highest CPU time (Table S7), being approximately 15 times higher than SVanalyzer and about 248 times higher than SURVIVOR. To evaluate their performance with 40 CPUs available, we conducted tests. However, the average usage was only 6 CPUs for svpop, whereas PanPop utilized 10.6 CPUs on average. Under these circumstances, the actual elapsed time of PanPop was 3 times faster than SVanalyzer and 11 times slower than SURVIVOR. Therefore, PanPop has the potential for even faster performance if more computing resources are available. In terms of memory usage, PanPop had a maximum memory usage of 1.7 GB, which is higher than other software tools. However, this level of memory usage is still manageable, even for a laptop.

For population SV merging of the 86 TGS and 1092 NGS dataset, PanPop also consumed the most CPU times among the tested software (Table S8). Additionally, as the data volume increasing, the memory usage also increases in both software. PanPop reached 17 Gb and 58 Gb in TGS and NGS analyses, respectively, which is also higher than the other software. Similarly, under the multiple thread process, the real elapsed time was greatly reduced.

In our advised manuscript, we added the running time, memory usage of all software, and emphasize the multithreaded characteristics of PanPop (Table S7-S8 and Supplementary notes 4, line 234-240).

5. The authors used custom parameters for Truvari, in particular `-mutimatch`? What was the intuition behind that? It would be informative to see the benchmarks using default parameters on a representative dataset (as a supplementary table).

Reply: The parameter ‘-multimatch’ allow one SV could match multiple SVs in the base dataset, which might slightly raise the number of true-positives. Without this parameter also

could be found in many papers (Popic et al. 2023 *Nature Methods*; Kolmogorov et al. 2023 *Nature Methods*), and may be more fair. So, we have deleted this parameter in the revised analyses. In addition, we also give the results under all parameters as default in source data to help readers fully consider all the pipelines.

Related references:

Popic, V., Rohlicek, C., Cunial, F. *et al.* Cue: a deep-learning framework for structural variant discovery and genotyping. *Nat Methods* **20**, 559–568 (2023). <https://doi.org/10.1038/s41592-023-01799-x>

Kolmogorov, M., Billingsley, K.J., Mastoras, M. *et al.* Scalable Nanopore sequencing of human genomes provides a comprehensive view of haplotype-resolved variation and methylation. *Nat Methods* **20**, 1483–1492 (2023). <https://doi.org/10.1038/s41592-023-01993-x>

6. When building a realign group, what happens to SVs that are within the clustering distance, but on different haplotypes? What happens in the case of overlapping SVs?

Reply: Apologies for the confusion in the previous description. During PART and PanPop process, we have taken into account the presence of structural variations (SVs) in different haplotypes for each individual. There are two scenarios that require different handling processes.

When SVs are nearby (within the clustering distance) heterozygous mutations in same haplotypes of an individual, we manually adjust the allele status of the second SV to simplify the situation. For example, if the original allele status is 0/1, we reverse it to 1/0 (or vice versa). This ensures that these SVs are divided into different haplotypes, thereby avoiding overlapping occurrences. This process is applied by using the ‘--first_merge’ parameter in the ‘scripts/realign.pl’ script, which is set as the default.

If there are two overlapping homozygous mutations in an individual, this situation should not actually occur and may be attributed to the incompleteness of SV callers. To handle this incompatibility, we introduced the ‘--skip_mut_at_same_pos’ parameter in the scripts/realign.pl script. There are three options for this parameter: If ‘skip_mut_at_same_pos’ is set to 0, no overlapping SVs are allowed, and the program will stop if overlapping SVs are detected. If ‘skip_mut_at_same_pos’ is set to 1, PanPop will skip the second overlapping SV. If ‘skip_mut_at_same_pos’ is set to 2, the program will attempt to fix the broken SV by reversing the allele status. If it cannot be fixed, only the overlapping bases will be discarded while the SVs will be retained. By default, skip_mut_at_same_pos was set to 3.

We have incorporated these explanations in the revised manuscript, specifically in lines 277-280, to provide a clear understanding of the process and its parameters.

7. The authors use the term “maximum missing rate” to benchmark the methods on a population

dataset. It needs to be clearly defined in the text. Is this a proportion of samples with missing genotypes?

Reply: You are correct. The ‘maximum missing rate’ means a proportion of samples with missing genotypes. We added this information in line 176-177 in the revised manuscript.

8. In these benchmarks, Assemblytics seems to be performing poorly. This is a fairly old method, and more recent assembly-based SV callers, such as SVIM-asm or dipdiff seem to have better performance. It might make sense to include one of them into the comparison.

Reply: Good suggestion. We have considered the two software for the comparison. While, dipdiff was designed for processing a pair of haplotypes, which were not suitable for our dataset. Hence, only SVIM-asm were added as a replacement of Assemblytics. The performance of SVIM-asm was amazing in *A. thaliana* simulation dataset, which generated a nearly full mark of SVs. But in complex situations as in HG002, the recall rate drops to about 70%. Still, this performance is greatly better than Assemblytics. All the results were added in the revised manuscript (Fig. 1, Fig. 3 and Line 110-111, 135-136)

9. Line 169: is the dataset Nanopore or PacBio?

Reply: This data was sequenced by HiFi. We have fixed this typo error in our revised manuscript (line 254).

10. Results from all tools that were used for evaluation should be made available via a public repository.

Reply: Good suggestion. We have followed your advice and uploaded all the raw results and intermediate files to figshare. This includes the data from various SV callers and all the SV mergers. The simulated dataset and VCFs of SV mergers for single individual SV merging are available at the following DOIs: 10.6084/m9.figshare.24657801 for the *A. thaliana* simulated dataset and 10.6084/m9.figshare.24658671 for the HG002 dataset. Additionally, the population-scale SV merge results and raw SV data for each TGS sample can be accessed at DOI 10.6084/m9.figshare.24659160. By providing access to these datasets on figshare, we ensure transparency and enable other researchers to replicate and verify our findings. This promotes open science and facilitates further exploration and analysis of the data. This part of information were added to data availability of our advised manuscript.

11. In supplementary information, the authors state that the software is free for academic, but not commercial use, but the license on github repository seems to be MIT.

Reply: Thanks for pointing out. We have changed the corresponding description into “PanPop is under MIT license”.

Reviewer #3 (Remarks to the Author):

In this study, the authors described a new method for merging structural variations, which was evaluated in detail using simulated dataset, individual dataset, and population-scale dataset. The study addressed an important issue in the field. The authors demonstrated that the method had commendable performance with potential applications. However, there are some concerns that require clarification.

Reply: We have diligently incorporated your feedback and made extensive revisions to the article. We hope this version will be more suitable for publication and satisfy you.

Major concerns:

1. The experiments mainly evaluated the performance of PanPop on simulated and *Arabidopsis thaliana* datasets. It is suggested to further validate PanPop on real data from other model animals, in order to further demonstrate that PanPop has good adaptability to different species.

Reply: We conducted evaluations using three real datasets, which included the human HG002 dataset for individual SV merging, as well as the *Arabidopsis* long and short reads datasets for both individual and population-scale merging. It is worth mentioning that model animals (e.g., human, mouse) typically have larger genome sizes compared to model plants (e.g., *Arabidopsis*, rice), which require significant computational resources for SV analyses. Due to limitations in our laboratory, we were only able to include the individual-scale SV merger test using the HG002 dataset and could not incorporate more datasets from model animals in this analysis. However, we would like to highlight a recently published study in Nature Communications (DOI: 10.1038/s41467-023-42029-4) that conducted a pan-genome analysis of yak using PanPop for SV merging, addressing this limitation. In fact, real plant datasets can be equally complex as animals, and even more so due to high repeat contents and variations. Therefore, PanPop is suitable and expected to deliver high performance in animal studies as well.

2. Unlike the non-repeat region, the breakpoint coordinates of SV obtained by different SV callers are inconsistent in the tandem repeat region. Inconsistent breakpoints affect SV merging. The authors need to take this into account in both simulated and real datasets. They may take different consideration for different regions.

Reply: We totally agree with that SVs located in repeat regions can exhibit inconsistent locations and variable SV lengths, which can pose challenges in the merging process. Within PanPop, incorporated two modules could address these issues. The "Realign-group" module gathers nearby SVs together, effectively handling the problem of inconsistent locations. Additionally, the "Local realign" module removes redundant sequences shared by SVs, enabling the extraction of multiple simple SVs from a larger and more complex SV. As a result, PanPop is capable of effectively handling these complex conditions.

Furthermore, we conducted tests to evaluate the performance of PanPop with SVs that

are entirely within repeat regions compared to those in non-repeat regions. After performing a chi-square test, we found no significant difference in the false positive (FP) or false negative (FN) counts between the two sets of SVs (See the following table). All relevant information regarding these tests has been included in the revised manuscript (Table S9, “SV merging of repeat and non-repeat regions” paragraph of Supplementary notes 6).

3. Line 51, MUSCLE is acknowledged as a reliable multiple sequence aligner. However, when dealing with numerous repeated sequences and significant variations in sequence length, misalignments may occur in the comparison results. How does the alignment accuracy affect the determination of biallelic and multiallelic SVs?

Reply: Thank you for this valuable comment. Alignment with the repeat sequences is a challenge for aligners and will give the variable SV positions. However, the exact position is not important in PanPop, we mainly focused on relative location. Also, benefited by the alignment method, PanPop were able to keep sequence information of SVs during merging process.

Here we used one example to clearly describe the process and the details were shown in the following figure (here name as Fig. 1). We assume that sequence of green part and yellow part were the tandem repeats with high similarity (Fig. 1a-e), and they both have a high similarity with Rep, Ins1 and Ins2 (Fig1. f). So, the SV of Rep, Ins1 and Ins2 could located in three locations, which is the start of green part, end of green part and end of yellow part. Although aligner may confuse for a ‘wrong’ position based on their alignment rules, but the sequence of Rep, Ins1 and Ins2 should be correct (Fig. 1c-e). These sequences will be thinned based on the aligned sequences, rather than the position, resulting in the splitting of these complex SVs into multiple simple SVs. Therefore, the thinning process greatly reduces the complexity of SVs and makes them easier for further alignment. As depicted in Figure 2 of the main text, PART will be employed multiple times, leading to more accurate results. Consequently, the full process yields the most accurate SVs.

Fig. 1. Diagram of three possible result of PART process. Raw SVs (a) processed by ‘rebuild sequences’ (b) could be alignment into three possible situations of c, d and e. Different colors represent different sequences or SVs (Del for deletion, Rep for repeat, Ins for insertion) for a and the first diagram of b, c and d. In upper three partition of diagram, thick lines with different color

indicated different sequences. The thin grey line indicated gap after alignment. The last two part of diagram of **c d** and **e** illustrate the allelic state of SVs after SV integration. **f**, collinear relationship between reference sequence, Rep, Ins1, and Ins2.

4. Give more details on the parameter settings and principles behind key parameters chosen for the PART algorithm. For example, lines 61-62 mention a threshold for the group length, but the rationale behind this choice is not clear. Is there a specific reason for selecting this threshold? If the group length threshold is modified, should the 20 bp difference also be adjusted?

Reply: Great suggestion. We have implemented an improved version of PanPop, and now the group length threshold can be easily modified in different analyses by adjusting the parameter 'sv2pav_merge_diff_threshold' in the config file. We conducted several tests with different thresholds, including 100bp, 50bp, 20bp, 10bp, and 5bp (Fig. S6). As the 'sv2pan_merge_diff_threshold' increased, the accuracy decreased (recall, precision, and F-score decreased), while the biallelic proportion increased. Therefore, users have the flexibility to set this parameter based on their specific goals. In this paper, we set the 'sv2pan_merge_diff_threshold' to a compromise value of 20bp as the default parameter. We have included all relevant information regarding these tests in the 'Group length threshold parameter' paragraph of the Supplementary notes 5 in the revised manuscript.

5. How are large SVs handled, such as deletions larger than one or two realignment groups? If a large SV spans multiple groups, is there a risk of it being split into multiple biallelic SVs?

Reply: It is not expected for a larger SV to cross two realignment groups. The range of realignment group were not fixed and continues to expand when there are nearby SVs. The end position of the realignment group is determined by the end position of each SV within the group, plus a certain number of bases. Therefore, larger SVs will be located within a single large group and will not be divided into different groups.

6. Please provide clear instructions on how to obtain the sequence during the "rebuild sequences" step in the pipeline. Is this step based on the BAM file or the VCF file? Not all VCF files from SV calling tools contain sequence information in the Info column.

Reply: We apologize for the previous inadequate description. We would like to provide a more detailed explanation of the "rebuild sequences" process.

All the necessary information for sequence reconstruction was obtained from the VCF files, which were preprocessed before use. For SV calls from long reads mapping callers, we utilized our script 'long_caller_parser.pl' to effectively extract and organize the relevant information from the Info column of the raw VCF files. This information includes the position of the SV types, the insertion sequences for insertions, and the SV lengths for deletions. With this extracted information, we were able to accurately reconstruct the sequences associated with the SVs.

In the case of assembly-based SV callers, the insertion sequence was determined based on the position of the query sequence. By utilizing this positional information, we successfully reconstructed the sequences related to the SVs.

To ensure clarity, we have included a detailed explanation of the "rebuild sequences" process in the revised manuscript, specifically at Line 79-82 and Supplementary notes 7.

7. Please evaluate the memory consumption and speed of the software.

Reply: Thanks for advising. We included this information in our advised manuscript and Table S7-S8 as advised. For further details, please refer to the response provided for reviewer #2 question 4.

Minor concerns:

1. The author may compare PanPop to sv-merger from the Icelandic population SV article (<https://doi.org/10.1038/s41588-021-00865-4>). (<https://github.com/DecodeGenetics/sv-merger>)

Reply: We have incorporated your suggested changes. The sv-merger tool shares similarities with SURVIVOR as both rely on the position of the reference genome. In our comparison, we evaluated the performance of sv-merger and other tools using a deletion dataset. While sv-merger demonstrated a high recall rate of 91.8% and a precision rate of 91.7%, PanPop exhibited even better performance (Fig. S4). We have cited the relevant paper and included all the pertinent results in the revised manuscript, specifically in the "Benchmark for sv-merger" section of Supplementary Notes 1, along with Figure S4.

2. Please expand the description of existing SV merging tools. Highlight their limitations in handling complex overlapping SVs.

Reply: Thanks for advising. We added brief summarizing of existing SV merges and highlight their limitations in the introduction (Line 31-33).

3. Please sharpen the conclusions by summarizing the key innovations over previous methods.

Reply: We have added a brief summarizing key innovations of PanPop in the conclusion section (Line 30-44).

4. How is the simulated SV dataset constructed, described in Methods.

Reply: We have added the brief description of simulated SVs in the revised Methods (Line 256-257) and the detail information in supplementary information documents (Line 142-156).

5. No other SV merger usage parameters, versions, and references are seen in Methods. Such as svtools, svimmer, SV-Pop, jasmine and other software.

Reply: We have included all the necessary details regarding the usage parameters, command lines, and versions in the revised supplementary information (lines 159-258). Additionally, we

have added the corresponding references in the revised main manuscript (line 105-106) to provide comprehensive information for readers.

6. In Figure 1b, both cuteSV and Sniffles exhibit higher F1-scores. Given that version 2 of Sniffles is capable of calling SVs at the population level, could you provide insights into the performance of Sniffles calls at the population level?

Reply: We have performed tests on Sniffles2 SV merging at the population level, and the results indicate that even when applying a 30% missing rate filter in Sniffles2, the recall and precision rates remained above 70%. This performance surpassed other available SV merging software options, although PanPop still demonstrated the highest performance (Fig. S5). All the pertinent details regarding these tests have been included in the revised manuscript, specifically in the "Benchmark for Sniffles2 for population SV" paragraph of the Supplementary Notes 3.

7. In Figures 1 and 4, the accuracy of individual SV callers appears generally low. To enhance clarity, it is recommended to present the results of two or three selected callers among the five.

Reply: We have conducted the analyses as per your suggestion. However, testing all possible combinations of two or three callers in different pipelines would require performing a total of $((C_5^2 + C_5^3) \times 4) = 80$ merging steps, which is computationally intensive and resource-consuming. Therefore, we decided to focus on extracting and benchmarking SVs that were supported by at least two or more software tools. The relevant results are presented in Figure 1, Figure 3, and Figure 4, and it was observed that the strategy of being supported by at least two SV callers yielded the best F1 score. This is mainly due to the precision rate increasing with the number of supporting callers, while the recall rate decreases significantly. Therefore, we recommend using support from two callers as the default selection, but we also provide an option to include more callers for specific use cases if the users have the specific purpose. We have added these descriptions in the revised manuscript (Line 113-118)

8. Lines 163-170 refer to the 9641 SVs of HG002, but not all of them are SV sets. Could you elaborate on how precision and sensitivity are evaluated based on this dataset?

Reply: Our benchmark dataset was obtained from the published paper of Zook et al. 2019 Nature Biotechnology, including the high confidence insertion and deletion callsets and the high confidence regions set. This dataset had been widely used as benchmarking performance based on SVs (e.g. Shafin et al. 2021 Nature Methods; Chen et al. 2023 Nature Communications; Cameron et al. 2019 Nature Communications). The total 9341 SVs that used in our benchmarking were the callsets SVs located in high confidence regions. When performing a benchmark, all SVs from an SV merger, high confidence callsets and high confidence regions were given to Truvari-benchmark. Only SVs located in the high confidence regions were used for benchmarking, which could treat as a region mask of SVs. The precision and sensitivity were calculated based on this part of SVs. To be more clearly, we added the

detail description in our advised manuscript at line 339-341.

Related references:

Zook et al. An open resource for accurately benchmarking small variant and reference calls. *Nat Biotechnol* 37, 561–566 (2019). <https://doi.org/10.1038/s41587-019-0074-6>

Cameron et al. A.T. Comprehensive evaluation and characterisation of short read general-purpose structural variant calling software. *Nat Commun* 10, 3240 (2019). <https://doi.org/10.1038/s41467-019-11146-4>

Shafin et al. Haplotype-aware variant calling with PEPPER-Margin-DeepVariant enables high accuracy in nanopore long-reads. *Nat Methods* 18, 1322–1332 (2021). <https://doi.org/10.1038/s41592-021-01299-w>

Chen et al. Deciphering the exact breakpoints of structural variations using long sequencing reads with DeBreak. *Nat Commun* 14, 283 (2023). <https://doi.org/10.1038/s41467-023-35996-1>

9. On line 176, it would be helpful to clearly specify the type of data.

Reply: Thanks for advising. The detailed dataset was described in Table S2. Totally 79 HiFi and 5 ONT sequenced were included. To be more clearly, we add a statistic of sequence type in our advised manuscript at line 261-263.

10. There are some errors in the manuscript, such as the F1-score value in Figure 4.

Reply: Thanks for pointing out and sorry for the typo. We fixed those errors in our advised manuscript.

11. Line 296: Updated reference to Jasmine, (Kirsche, M., Prabhu, G., Sherman, R. et al. Jasmine and Iris: population-scale structural variant comparison and analysis. *Nat Methods* 20, 408–417 (2023). <https://doi.org/10.1038/s41592-022-01753-3>)

Reply: Done.

Reviewers' Comments:

Reviewer #1:

Remarks to the Author:

I am glad to find the SV data have been made available for download. Also the description of the PART module is now more clear. Regarding the revised Table S2, its name (Table S2. Data source of 84 long sequenced *A. thaliana*) should be also modified - according to the revised manuscript there are 86 samples (plus Col-PEK?). Also, the number of SVs for the last sample in the table is 0.0. Is this a typo? According to the Methods section there should be no samples with the number of SV < 100? Nevertheless, these errors are not essential and they do not affect the results of the PanPop validation. I appreciate all improvements added in this revised version and have no further comments.

Reviewer #2:

Remarks to the Author:

I thank the authors for answering all my requests in detail, I have no further questions.

Reviewer #3:

Remarks to the Author:

In this revised version, the authors have satisfactorily addressed all of my concerns. The manuscript has significantly improved, rendering it clearer and more refined. I no longer have any other concerns.

Reviewer #1 (Remarks to the Author):

I am glad to find the SV data have been made available for download. Also the description of the PART module is now more clear. Regarding the revised Table S2, its name (Table S2. Data source of 84 long sequenced *A. thaliana*) should be also modified - according to the revised manuscript there are 86 samples (plus Col-PEK?). Also, the number of SVs for the last sample in the table is 0.0. Is this a typo? According to the Methods section there should be no samples with the number of SV < 100? Nevertheless, these errors are not essential and they do not affect the results of the PanPop validation. I appreciate all improvements added in this revised version and have no further comments.

Reply: Thank you for your diligent review and approval of our manuscript. We sincerely apologize for the presence of typos and any inconvenience caused. In response to your feedback, we have made the necessary corrections. Firstly, we have updated the name of Table S2 to accurately reflect its content as "Data source of 86 long sequenced *A. thaliana* and the Col-PEK reference." Additionally, we have rectified the typo in the last value of Table S2, which now stands corrected as 8245. We greatly appreciate your attention to detail and assistance in improving the accuracy of our manuscript.